# Pre-training via Paraphrasing

**Mike Lewis**  **Marjan Ghazvininejad**  **Gargi Ghosh**

**Armen Aghajanyan**  **Sida Wang**  **Luke Zettlemoyer**

Facebook AI
mikelewis@fb.com

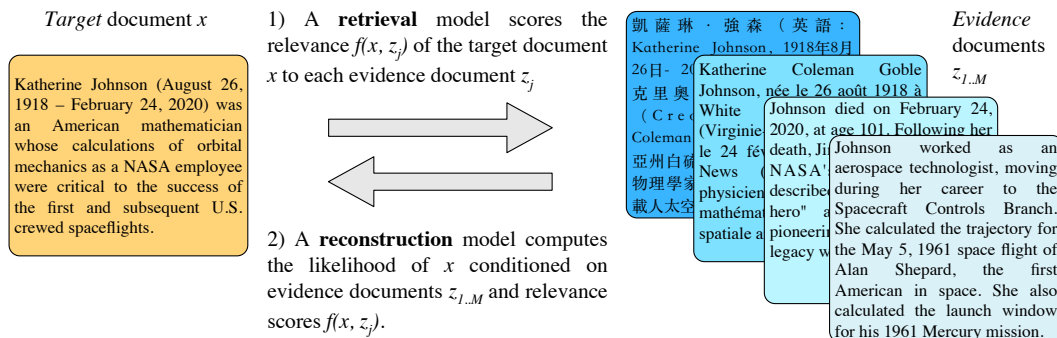

Figure 1: Pre-training via Paraphrasing: a retrieval model maps a document to a set of related documents, which a reconstruction model paraphrases to maximize the likelihood of the original. Example text adapted from `https://{en,es,de,it,fr,zh}.wikipedia.org/wiki/Katherine_Johnson`

## Abstract

We introduce MARGE, a pre-trained sequence-to-sequence model learned with an unsupervised multi-lingual multi-document paraphrasing objective. MARGE provides an alternative to the dominant masked language modeling paradigm, where we self-supervise the *reconstruction* of target text by *retrieving* a set of related texts (in many languages) and conditioning on them to maximize the likelihood of generating the original. We show it is possible to jointly learn to do retrieval and reconstruction, given only a random initialization. The objective noisily captures aspects of paraphrase, translation, multi-document summarization, and information retrieval, allowing for strong zero-shot performance on several tasks. For example, with no additional task-specific training we achieve BLEU scores of up to 35.8 for document translation. We further show that fine-tuning gives strong performance on a range of discriminative and generative tasks in many languages, making MARGE the most generally applicable pre-training method to date.

## 1   Introduction

Variations on masked language models (MLMs) [Devlin et al., 2019, Liu et al., 2019, Yang et al., 2019b, Conneau et al., 2019, Lewis et al., 2019a, Raffel et al., 2019, Clark et al., 2020] provide highly effective self supervision for pre-training by removing and then reconstructing parts of an input text. In this paper, we present the first viable pretraining alternative to MLMs; self supervision is instead provided by learning to paraphrase collections of related documents in many languages.

More specifically, we introduce MARGE, a **M**ultilingual **A**utoencoder that **R**etrieves and **Ge**nerates. We train MARGE by self-supervising the *reconstruction* of target text by first *retrieving* a set of related texts (in many languages) and then conditioning on them to maximize the likelihood of generating the original. We pre-train a multi-source sequence to sequence model that separately encodes each retrieved document and decodes the target, piecing together and translating content from the appropriate inputs as needed to provide the best reconstruction possible. The retrieval model scores are used to bias the cross attention to the most relevant retrieved documents, allowing the retrieval model to be trained jointly from the reconstruction loss.

Our approach can be viewed as a new type of denoising auto-encoder where the noise comes from the retrieval step and is much more diverse than masking; retrieved documents may have little lexical overlap with the target, and may not even be in the same language, but should communicate the same underlying information. The pre-training task emphasizes paraphrasing and reduces the amount of encyclopedic knowledge the model must memorize. The set of retrieved documents and relevance scores are an autoencoder bottleneck from which the input must be reconstructed. MARGE is related to recent work that learns to do retrieval as part of the end task model, for example to find evidence documents in open domain question answering [Guu et al., 2020, Lewis et al., 2020]. This leads to a more challenging retrieval problem that, unlike ours, requires a separate pre-training phase.

Overall, our pre-trained models capture elements of traditional paraphrasing, translation, multi-document summarization, and information retrieval tasks.[1] This enables effective zero-shot learning; with no fine-tuning we achieve BLEU scores of up to 35.8 for document translation, and outperform strong baselines for cross-lingual transfer in summarization — providing a step towards pre-trained models that can perform any task with little or no fine-tuning. With fine-tuning, we achieve competitive performance with masked language models on a range of discriminative and generative tasks in many languages, making MARGE the most generally applicable pre-training method to date.

## 2 Model

### 2.1 Overview

During pre-training, the input to the model is a batch of evidence documents[2] $z_{1..M}$ and target documents $x_{1..N}$. The model is trained to maximize the likelihood of the targets, conditioned on the evidence documents, and the relevance of each evidence document to each target:

- The model first computes a relevance score $f(x_i, z_j)$ between every pair of documents $x_i$ and $z_j$, by embedding each document and computing their cosine similarities (§2.2).

- The model then computes the likelihood of reconstructing each $x_i$ conditioned on $z_{1..M}$ and each $f(x_i, \cdot)$, using a modified seq2seq model. The similarity score encourages the model to attend more to relevant evidence documents. Backpropagating the reconstruction loss therefore improves both the sequence-to-sequence model and the relevance model (§2.3).

- We construct batches so that evidence documents are relevant to the targets, using the relevance model for retrieval (§2.4).

Training this model is a chicken-and-egg problem. The reconstruction and relevance models cannot be effectively updated if the batches do not contain relevant evidence documents, but batch construction relies on a relevance model. However, we found that, in practice, the model is able to learn from a random initialization, which effectively provides a type of hashing of random features for each word.

## 2.2 Relevance Scores

To learn the relevance scores $f(x_i, z_j)$ for a pair of documents, we train a document encoder $g$ that maps a list of tokens to a fixed size representation. We apply the same encoder to both the target and evidence document, and take the cosine similarity between their representations:

$$f(x, z) = \begin{cases} \frac{g(x) \cdot g(z)}{\|g(x)\| \|g(z)\|} & \text{if } x \neq z \\ -\infty & \text{otherwise} \end{cases} \tag{1}$$

This function is used in the reconstruction model (§2.3), and trained by the reconstruction loss. It is also used to construct batches of relevant documents (§2.4).

Using the same encoder for both the target and evidence documents allows even random models to compute meaningful similarity functions, as documents with high lexical overlap are more likely to be projected to more similar representations [Wieting and Kiela, 2019]. This is crucial at initialization.

We encode documents by taking the representation of the first token from the top of a 4-layer Transformer [Vaswani et al., 2017]. We share parameters with the first four layers of the reconstruction-model encoder, which saves computation and allows multitask learning.

## 2.3 Reconstruction Model

Given a set of evidence documents $z_{1..M}$ and similarity scores $f(x_i, z_j)$, the reconstruction model computes the likelihood of target document $x_i$.

$$L_\theta = - \sum_i \log p_\theta(x_i | z_{1..M}, f(x_i, z_1), \dots, f(x_i, z_M)) \tag{2}$$

This provides an auto-encoder loss where the reconstruction of document $x_i$ is indirectly conditioned on $x_i$, but with an intermediate bottleneck provided by the retrieved documents and relevance scores.

First, the input documents are encoded individually with a bidirectional Transformer, and then the resulting embeddings are concatenated. The similarity score is used to bias the cross-attention from the decoder to the encoder, so that the decoder will pay more attention to more relevant evidence documents. Using more relevant evidence documents will improve the likelihood of reconstructing $x_i$, so gradient descent on (2) will improve the quality of the similarity scores.

Standard Transformer sequence-to-sequence models [Vaswani et al., 2017] compute a matrix of cross-attention probabilities between all elements of target document $x_i$ and evidence document $z_j$:

$$\alpha = softmax_{z_j}(Q^{lh}(x_i)K^{lh}(z_j)) \in \mathbb{R}^{|x_i| \times |z_j|} \tag{3}$$

where $Q^{lh}$ and $K^{lh}$ compute query and key representations for layer $l$ and head $h$, and $softmax_{z_j}$ denotes a softmax normalised over elements of $z_j$.

We instead compute cross attention over a set of evidence documents $z_{1..M}$, biasing the attention scores with the document relevant score from (1):

$$\alpha = softmax_{z_{1..M}}(Q^{lh}(x_i)K^{lh}(z_{1..M}) + \beta f(x_i, z_j)) \in \mathbb{R}^{|x_i| \times \sum_j |z_j|} \tag{4}$$

where $\beta$ is a trainable scalar parameter that weights the importance of the document similarity score.

Guu et al. [2020] propose a related approach in which the likelihood of a target $x$ is calculated by marginalizing out latent documents $z$: $p(x) = \sum_j p(x|z_j)p(z_j)$. Our attention-like mechanism is (1) more expressive, because it can pay complete attention to a token from one document at one timestep and a token from another document at another timestep, and (2) more efficient because $p(x|z)$ is not computed separately for each $z_j$. However, our method does not allow attention from $z$ to $x$.

## 2.4 Batch Construction

Batches are constructed to create evidence document sets $z_{1..M}$ that give useful information for reconstructing target documents $x_{1..N}$, as detailed in this section. Overall, we divide the data into

*shards* of related documents. Periodically, we compute the similarities between pairs of documents within each shard, using the relevance model, and apply a threshold to keep the strongest connections. The final batches are constructed to maximize connectivity between evidence and target documents.

**Document similarity**   We compute document similarity in the same way as §2.2. All documents $x$ are encoded as a vector $g(x) \in \mathbb{R}^d$, and then all pair-wise similarities between documents are computed with a single matrix multiplication.

**Data Sharding**   We use simple heuristic constraints to divide documents into related shards, to improve both the accuracy and efficiency of retrieval. Specifically, for news text, documents are in the same shard iff they were published on the same date. For Wikipedia, we split articles into chunks of length 512. We create 1000 shards, where all chunks from the same article, or the equivalent article in another language, are in the same shard (otherwise dividing chunks randomly). Shards typically contain 50-250k entries.

**Indexing**   While we backpropagate through the relevance model in (4), the construction of the batch itself is inherently non-differentiable. For convenience we perform the nearest neighbour search offline. Every 10k model updates, we sample a set of shards of documents. For each shard, we compute $f(x, z)$ for every pair of target and evidence documents, using the current relevance model.

**Thresholding**   We select which documents are sufficiently related by taking the top $k$ most similar document pairs across all pairs in the shard. Some targets may have no sufficiently relevant evidence documents, and are unused until the shard is re-indexed with an updated relevance model.

**Batching**   We aim to construct batches containing clusters of related target and evidence documents, to maximize available information for reconstructing each target. The output from the thresholding step is a bipartite graph of evidence and target documents with edges between them. A batch is a subgraph, and we perform a small local search to find subgraphs maximizing the sum of the weights of all edges in the subgraph. To encourage the model to build multilingual batches, edges where the evidence and target are in different languages are given weight 100, and other edges have weight 1. To create batches, we iterate over seed evidence documents $x_i$ with an edge to at least one evidence document. We then greedily add evidence and target documents to the batch to maximize the sum of the weights of edges, until the maximum number of tokens that can fit in GPU memory is reached.

# 3   Training

**Architecture**   We use a Transformer model [Vaswani et al., 2017]. The encoder consists of 12 Transformer layers of dimension 1024, with feedforward layers of size 4096. Recent work showed that large models train more efficiently [Li et al., 2020, Kaplan et al., 2020]. The decoder is similar to the encoder, but we increase the size of the feed-forward layers in the Transformer decoder to 16536. We also add 4 additional Transformer layers to the base of the decoder with only self-attention and feedforward layers of size 4096, which allows words in the target to contextualize locally before the more expensive cross-attention and feed-forward layers. We focus on scaling up the decoder, because it has access to more information than the encoder (which sees only evidence documents). In total, the model contains roughly 960M parameters. For the relevance model, we use the first 4 layers of the encoder, and take the documents representation from the beginning-of-sentence token.

**Pre-training**   During pre-training, workers process sub-batches containing an average of 2 evidence documents and 2 target documents, and accumulate gradients across workers. Using a multilingual version of the CC-NEWS corpus [Liu et al., 2019], we train initially using the with 64 workers for 450k steps (linearly annealing the learning rate from 1e-04 to 0 with 10k warmup steps), and then continue training with 2048 workers with 550k steps (annealing the learning rate from 2e-04 to 0).[3] We refer to this model as MARGE-NEWS. To explore domain effects, we further pre-train for 100k steps on Wikipedia data, annealing the learning rate from 1e-04 to 0, and refer to the resulting model as MARGE. We rebuild the index every 10k updates. We set retrieval thresholds such that we take on average 4 monolingual and 4 crosslingual links per target document.

| | #Parameters | #Languages | Pretraining task | Pretraining GPU Days (estimated) | Pretraining Data (GB; estimated) |
|---|---|---|---|---|---|
| mBERT | 172M | 104 | MLM | Unknown | 60 |
| XLM | 570M | 100 | MLM | 640 | 60 |
| XLM-R | 550M | 100 | MLM | 27000 | 2394 |
| MMTE | 192M | 100 | Translation | Unknown | Unknown |
| mBART | 680M | 25 | seq2seq MLM | 4500 | 1370 |
| MARGE | 963M | 26 | Retrieval+Reconstruction | 4700 | 206 |

Table 1: Comparison models: MARGE is pre-trained on a scale between XLM and XLM-R.

| | IWSLT2017 | | | | | WMT19 |
|---|---|---|---|---|---|---|
| | ar | de | fr | ja | zh | de |
| Into English | 26.8 | 28.5 | 34.3 | 12.6 | 19.9 | 35.8 |
| From English | 12.9 | 14.4 | 25.5 | 10.7 | 12.9 | 13.4 |

| | | | | Target | | |
|---|---|---|---|---|---|---|
| | | de | en | it | nl | ro |
| Source | de | - | 30.6 | 14.0 | 14.8 | 11.6 |
| | en | 18.8 | - | 14.3 | 15.0 | 14.0 |
| | it | 14.0 | 31.7 | - | 11.3 | 12.7 |
| | nl | 14.3 | 27.5 | 12.6 | - | 9.3 |
| | ro | 14.3 | 32.8 | 14.4 | 9.8 | - |

Table 2: **Zero-shot unsupervised document level machine translation** BLEU scores using the pre-trained model, with no fine-tuning or special constraints on generation. Performance varies considerably across languages, but is non-trivial with even distantly related languages.

**Data Pre-processing** We de-duplicate the data, and identify languages using FastText [Joulin et al., 2016]. We select documents published in 26 different languages (based on their prevalence in downstream tasks), summarized in the Appendix. We divide documents into chunks of length 512. We allow all chunks to be evidence documents. For the news domain, we only allow the first chunk in each document to be used as a target, which we found improved performance during development. We prepend a language identifier token as the first decoder input, to control the output language.

**Fine-tuning** For fine-tuning, we use a similar procedure to Lewis et al. [2019a]. For generation problems, such as translation and summarization, the task input is fed into the encoder, and the output is generated by the decoder. For classification problems the task input is fed into both the encoder and decoder, and a representation is used from the decoder's final layer hidden state. For zero-shot transfer experiments, we freeze word embeddings and the first 4 decoder layers.

## 4 Experiments

As a multi-lingual sequence-to-sequence model, MARGE is applicable to a very broad range of tasks. We focus on multi-lingual tasks with elements of retrieval, document comprehension, and document generation, because they are the most directly related to our pre-training.

Table 1 lists the strongest available multilingual pre-trained models, along with relevant model statistics. We compare performance to published numbers for these models.

### 4.1 Cross-lingual Sentence Retrieval

Our pre-training task requires the model to retrieve similar texts, which may be in different languages. As an extrinsic evaluation of this functionality, we study cross-lingual sentence retrieval, in which a model must identify the correct translation of a sentence from a set of distractors. We report performance on BUCC2018 [Zweigenbaum et al., 2018] and Tatoeba [Artetxe and Schwenk, 2019].

We follow the setup of Hu et al. [2020], using no fine-tuning. As a document representation, we use the average embedding of the fifth encoder layer (tuned on BUCC development data).

On BUCC (Table 3), MARGE outperforms other unsupervised models by almost 10 points. On Tatoeba (see Appendix), there is significant variation across languages, but overall MARGE performs comparably to XLM-R and significantly better than other pre-trained models. Better results have been achieved on both tasks using labeled bitext for training [Artetxe and Schwenk, 2019], but our results suggest that our pre-training objective learns an effective cross-lingual retrieval function.

| | de | fr | ru | zh | avg |
|---|---|---|---|---|---|
| mBERT | 62.5 | 62.6 | 51.8 | 50.0 | 56.7 |
| MMTE | 67.9 | 63.9 | 54.3 | 53.3 | 59.8 |
| XLM | 56.3 | 63.9 | 60.6 | 46.6 | 56.8 |
| XLM-R | 67.5 | 66.5 | 73.5 | 56.7 | 66.0 |
| MARGE | **78.8** | **75.9** | **77.3** | **71.6** | **75.9** |

Table 3: Unsupervised Sentence Retrieval results on BUCC. MARGE outperforms other unsupervised models.

| | en-de | zh-en |
|---|---|---|
| Random Initialization | 7.7 | 3.2 |
| HAN [Miculicich et al., 2018] | - | 24.0 |
| mBART (sentence) | 38.0 | 28.4 |
| mBART (document) | 38.5 | **29.6** |
| MARGE | **39.2** | 28.4 |

Table 4: Supervised document-level machine translation. Comparison results are from Liu et al. [2020]. MARGE performs similarly to mBART.

## 4.2 Document-Level Machine Translation

During pre-training, the model can retrieve evidence documents in different languages than the target—in contrast to mBERT, XLM and mBART where instances are monolingual. We explore how well this pre-training approach learns to translate. We focus on document level translation tasks, and report document-level BLEU scores.[4] Following Liu et al. [2020], we segment documents into chunks of 512 tokens for training and generation, and then concatenate chunks of the same document.

**Zero-Shot Unsupervised Document Translation**  Translation offers a direct measure of how well the pre-trained model encoder and decoder work for different languages, and the extent to which the interface between them is language independent. Therefore, in contrast to prior work on unsupervised translation, we do not further fine-tune the model with iterative back-translation [Lample et al., 2017, Artetxe et al., 2017], or bitext in other language pairs [Johnson et al., 2017, Liu et al., 2020].

We measure both translation into English, which compares encoder performance for other languages, and translation out of English, which measures the decoder performance. Generation hyperparameters were minimally tuned on German/English development, and are shared across all translation pairs. We use a beam of size 6 and block repeated n-grams of length 8 [Fan et al., 2017].

Results are shown in Table 2. Performance varies considerably by language, but reaches 35.8 for German to English, which is the highest score we are aware of for system trained with no bitext. Performance is also strong for some languages using different scripts, such as Arabic to English. However, some languages work less well, notably Japanese. Generating non-English languages proves harder in all cases, particularly those with non-Latin alphabets, but English to French works well. Future work should explore up-sampling rarer languages during pre-training.

Qualitatively, we note that the translations are often good but less literal translations than the reference. This may cause BLEU scores to underestimate performance. It is likely that unsupervised performance could be further improved using iterative back-translation using MARGE as an initialization, but we focus here on examining the pre-trained model directly.

**Supervised Document Translation**  We also evaluate how well our models can be fine-tuned for translation using labeled bitext. To compare with mBART, we use the same English-German and Chinese-English document translation tasks from WMT19 and IWSLT2015. Table 4 show that MARGE and mBART perform similarly, with MARGE performing better on English-German and mBART on Chinese-English. Both outperform baselines by a wide margin.

## 4.3 Summarization

We evaluate monolingual sequence-to-sequence generation performance on text summarization tasks. We use the MLSum dataset [Scialom et al., 2020] to compare performance in several languages.

Results are shown in Table 5. MARGE outperforms an extractive mBERT model—the extractive oracle performance suggests that extractive models are very competitive on this dataset—and a seq2seq model without pre-training. In some cases, training one model on all languages (train all) improves results. Finally, we explore zero-shot summarization, where the model is trained on all

| | | MLSum | | | | | |
|---|---|---|---|---|---|---|---|
| Model | Setting | de | es | fr | ru | tr | avg |
| Extractive Oracle | Oracle | 52.30 | 35.78 | 37.69 | 29.80 | 45.78 | 29.81 |
| Lead 3 | Deterministic | 33.09 | 13.70 | 19.69 | 5.94 | 28.90 | 13.65 |
| Pointer-Generator | Train One | 35.08 | 17.67 | 23.58 | 5.71 | 32.59 | 15.91 |
| M-BERT | Train One | 42.01 | 20.44 | 25.09 | 9.48 | 32.94 | 17.59 |
| MARGE-NEWS | Zero-shot Transfer | 30.01 | 17.81 | 19.39 | 8.67 | 29.39 | 15.05 |
| MARGE-NEWS | Train One | 42.60 | 22.31 | **25.91** | 10.85 | **36.09** | 19.03 |
| MARGE | Train All | 42.70 | 22.27 | 25.78 | 10.85 | 35.47 | 18.87 |
| MARGE-NEWS | Train All | **42.77** | **22.72** | 25.79 | **11.03** | 35.90 | **19.09** |

Table 5: ROUGE-L scores on MLSum. MARGE generates abstractive summaries that outperform an extractive mBERT model. We also demonstrate zero-shot transfer learning, where the model is trained only on languages it is not trained on, and results from training on all languages.

| | en | ar | de | es | hi | vi | zh | avg | en | de | es | fr | ja | ko | zh | avg |
|---|---|---|---|---|---|---|---|---|---|---|---|---|---|---|---|---|
| mBERT | 80.2 | 52.3 | 59.0 | 67.4 | 50.2 | 61.2 | 59.6 | 61.4 | 94.0 | 85.7 | 87.4 | 87.0 | 73.0 | 69.6 | 77.0 | 81.9 |
| MMTE | 78.5 | 56.1 | 58.4 | 64.9 | 46.2 | 59.4 | 58.3 | 60.3 | 93.1 | 85.1 | 87.2 | 86.9 | 72.0 | 69.2 | 75.9 | 81.3 |
| XLM | 68.6 | 42.5 | 50.8 | 54.7 | 34.4 | 48.3 | 40.5 | 48.5 | 94.0 | 85.9 | 88.3 | 87.4 | 69.3 | 64.8 | 76.5 | 80.9 |
| XLM-R | 83.5 | **66.6** | **70.1** | **74.1** | **70.6** | **74.0** | 62.1 | **71.6** | **94.7** | **89.7** | 90.1 | 90.4 | 78.7 | **79.0** | 82.3 | 86.4 |
| MARGE | **83.7** | 64.5 | 68.7 | 73.4 | 67.2 | 71.5 | **67.8** | 71.0 | **94.7** | 89.4 | **91.6** | 90.9 | 78.9 | 77.7 | **82.5** | 86.5 |

(a) F1 scores on the MLQA question answering task.      (b) Paraphrasing accuracy on PAWS-X.

Table 6: Cross-lingual transfer: models are trained on English (en) and tested on other languages. MARGE performs competitively with XLM-R, with 20% of the pre-training compute.

languages except the test language—this model outperforms a strong lead-3 baseline, and even a supervised pointer-generator model on Spanish and Russian. On this domain, we achieve better results with MARGE-NEWS, a version of the model trained only on news.

### 4.4 Paraphrasing

We measure how well our pre-training learns paraphrasing on the PAWS-X paraphrase detection dataset [Yang et al., 2019a]. The task is to determine whether two sentences are paraphrases; examples were constructed adversarially to have high lexical overlap. Models are trained on English, and we test zero-shot transfer to other languages. MARGE edges out a new state of the art (Table 6b).

### 4.5 Question Answering

Question answering offers another document level reasoning task that is easily posed in many languages. We use the MLQA dataset [Lewis et al., 2019b], in which models are trained on the English SQuAD dataset [Rajpurkar et al., 2016] and then tested in other languages. Results in Table 6a show that MARGE achieves competitive performance with XLM-R, setting the state of the art for Chinese, and outperforms other models by a wide margin.

## 5 Analysis

**What does the reconstruction model learn?** To build intuitions about what the reconstruction model learns, we examine model outputs for inputs in different languages on the same topic (Table 9). Even for a fixed topic, the model output varies significantly with the input, showing that it is not simply memorizing text. Almost all facts in the outputs are supported by the input, with few hallucinations—suggesting pre-training has taught the model to translate and paraphrase information from its source, rather than memorize facts in its parameters. However, the outputs are not literal translations—in particular, some important facts from the source are not expressed in the output. The model was not trained on literal translations, so it is perhaps surprising that the output is so closely aligned to the input. Translations may represent a mode of a diverse distribution over paraphrases.

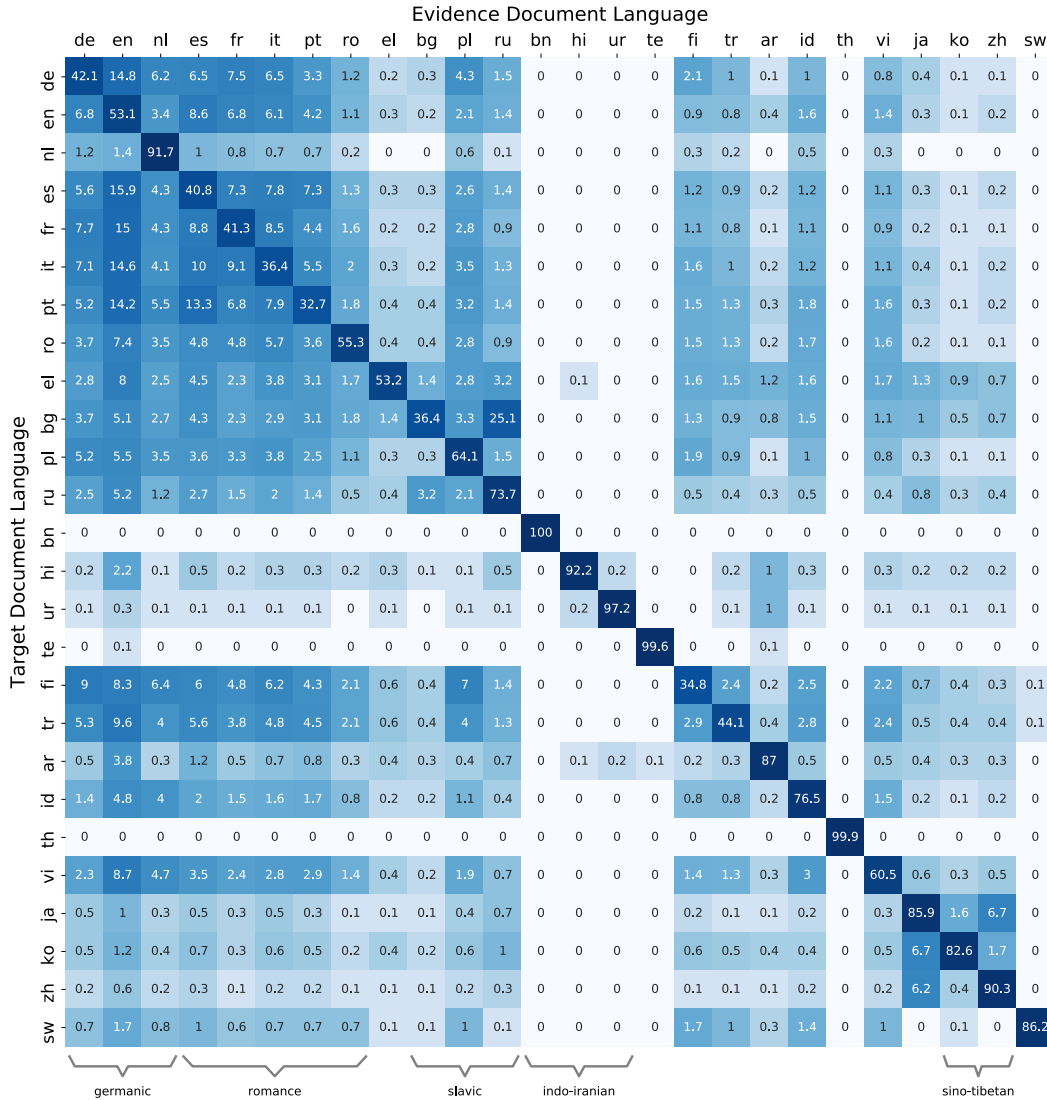

Figure 2: Percentage of retrieved links to documents in target languages (y-axis) from evidence documents in different source languages (x-axis) on Wikipedia.

**What does the retrieval model learn?**  Figure 2 shows statistics of the retrieval model. Differences across languages are due to many factors, including the frequency of languages in the corpus, and how related languages are to each other. Our pre-training also introduces feedback loops, because if the reconstruction model is unable to translate between two languages, it may train the retrieval model that documents in these languages are less relevant to each other.

All languages retrieve the highest proportion of documents within their own language (represented by the diagonal), but otherwise the retrieved documents tend to be distributed over a number of other languages. There tend to be closer affinities between geographically or linguistically related languages, such as Bulgarian and Russian, or Chinese and Japanese. For some languages, the model fails to retrieve many documents in other languages—particularly Indo-Iranian languages, and those which are the only example of their language family we include (such as Telugu and Thai). For these cases, the pre-training reduces to independent updates for each language, as in mBART and mBERT.

**Discussion**  Overall, MARGE shows strong performance on a wider range of tasks than any previous pre-trained models, and is effective at discriminative and generative tasks in many languages. Results are competitive with less general models, even XLM-R, which was trained with significantly

higher pre-training resources. The pre-training task is more closely related to downstream tasks than masked language modeling, allowing pre-trained models to achieve BLEU scores as high as 35.8 for translation. MARGE also broadens the range of known effective pre-training tasks beyond MLMs, which we hope will lead to further exploration and understanding of pre-training objectives.

However, there are several limitations that future work should address. We pre-trained on news and Wikipedia, where simple metadata can be used to constrain the similarity search, improving efficiency and accuracy. Broadening the domains may require approximate nearest neighbor search [Johnson et al., 2019]. Learning the retrieval model requires batch sizes greater than one, so model-parallel training would be required to train significantly larger models.

## 6    Related Work

**NLP pre-training**    Since BERT [Devlin et al., 2019], pre-training for NLP has been dominated by variants of masked language models. For example, Yang et al. [2019b] predicts the masked tokens auto-regressively, Dong et al. [2019] multitasks MLM and language modeling objectives, Clark et al. [2020] trains a discriminator to classify the correctness of MLM samples, and Lewis et al. [2019a] and Raffel et al. [2019] use seq2seq models with masked inputs. MARGE departs significantly from these objectives in that the inputs during pre-training are complete, uncorrupted text.

**Bitext Mining**    Recent work has shown impressive results on machine translation through bitext mining [Schwenk et al., 2019], in which a retrieval model is used to search for parallel sentences in a large multilingual corpus, which are then used as training data for a machine translation model. A key conceptual difference is that literal bitext is not optimal for our approach, as we hope to learn linguistic information by training on noisy document-level paraphrases. We also learn to retrieve and translate with no manually translated sentences, unlike existing bitext mining methods.

**Cross-lingual Learning**    Several works attempt to pre-train language-independent representations. McCann et al. [2017] and Siddhant et al. [2019] pre-train on translation tasks. Better resuts are achieved using MLMs on the concatenation of monolingual corpora, relying on parameter sharing to learn cross-lingual representations [Lample and Conneau, 2019, Conneau et al., 2019, Liu et al., 2020]. We instead pre-train on loose cross-lingual paraphrases.

**Language Models with Retrieval**    Several recent papers have shown that word prediction can be improved by retrieving relevant evidence documents. Guu et al. [2020] and Lewis et al. [2020] improve MLMs and text generation by learning to retrieve relevant evidence documents. Guu et al. [2018] perform language modeling by retrieving and editing sentences. kNN-LM [Khandelwal et al., 2019] shows that language models can be improved with retrieving from the training set, by interpolating a language model with a nearest neighbor classifier. In contrast, we learn retrieval during training but do not require it for inference. Perhaps most relevantly, Liu et al. [2018] generate Wikipedia articles conditioned on a set of evidence documents.

## 7    Conclusion

We introduced a new approach to pre-training models for natural language understanding and generation, by using retrieved documents to reconstruct the original. MARGE exhibits strong performance on a range of discriminative and generative tasks in many languages, both with and without fine-tuning. These results establish MARGE as a viable alternative to masked language modeling and provide a step towards pre-trained models that can perform any task with little or no fine-tuning. Future work should scale MARGE to more domains and languages.

## Broader Impact

This work has a broad scope, covering discriminative and generative tasks in many languages. As such, the broader impact is similar to that of the field of NLP; there exist many potential good and bad applications. The pre-training method is likely to capture and potentially amplify any biases found in the pre-training corpus. Our work learns to translate languages in an unsupervised way, which could

be used to bring NLP to more languages, but could also potentially introduce more translation errors that supervised methods.

## Acknowledgments

We would like to thank the reviewers and meta-reviewer for their thoughtful comments and suggestions.

## Footnotes

[1]Masked language models, in contrast, are less directly related to target fine tuning tasks and significant ongoing research focuses on understanding why they work so well, see Rogers et al. [2020] for a survey.

[2]We use *document* to refer to contiguous chunks of text up to maximum length (here, 512 tokens).

[3]Initially training with a smaller learning rate reduced instability with an untrained retrieval model.

[4]All sentences in a document are concatenated prior to calculating BLEU, using SacreBLEU [Post, 2018].

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
