[Supplementary Material]

# A   Additional Results

|        | ar   | bg   | bn   | de   | el   | es   | fi   | fr   | hi   | id   | it   | ja   |
|--------|------|------|------|------|------|------|------|------|------|------|------|------|
| XLM-R  | 47.5 | **71.6** | **43.0** | 88.8 | **61.8** | 75.7 | **71.6** | 73.7 | **72.2** | **77.0** | 68.3 | **60.6** |
| MARGE  | **49.9** | 70.5 | 16.9 | **88.9** | 57.2 | **82.9** | 55.8 | **77.0** | 67.1 | 73.8 | **76.5** | 60.1 |

|        | ko   | nl   | pt   | ru   | sw   | te   | th   | tr   | ur   | vi   | zh   |
|--------|------|------|------|------|------|------|------|------|------|------|------|
| XLM-R  | **61.4** | 80.8 | 82.2 | 74.1 | 20.3 | **35.9** | 29.4 | **65.7** | 24.3 | 74.7 | 68.3 |
| MARGE  | 50.6 | **84.3** | **84.8** | **78.7** | **22.8** | 16.2 | **38.0** | 63.2 | **41.9** | **77.3** | **77.2** |

Table 7: **Tatoeba** zero-shot sentence retrieval results. MARGE performs comparably to XLM-R, but with significant variation across languages. We only show results for languages in all model's pre-training data.

# B   Pre-training Data

| Language   | Code | Language Family | CCNews   | Wikipedia |
|------------|------|-----------------|----------|-----------|
| Arabic     | ar   | Afro-Asiatic    | 2416996  | 747891    |
| Bulgarian  | bg   | Slavic          | 496023   | 297989    |
| Bengali    | bn   | Indo-Iranian    | 741      | 134560    |
| German     | de   | Germanic        | 13320055 | 2735591   |
| Greek      | el   | Hellenic        | 1793198  | 317780    |
| English    | en   | Germanic        | 57061325 | 6372976   |
| Spanish    | es   | Romance         | 16990991 | 2111406   |
| Finnish    | fi   | Uralic          | 471029   | 496988    |
| French     | fr   | Romance         | 7281926  | 2749382   |
| Hindi      | hi   | Indo-Iranian    | 1907850  | 124816    |
| Indonesian | id   | Austronesian    | 1295060  | 435599    |
| Italian    | it   | Romance         | 6865752  | 1776998   |
| Japanese   | ja   | Japonic         | 458675   | 1311915   |
| Korean     | ko   | Sino-Tibetan    | 1241560  | 442675    |
| Dutch      | nl   | Germanic        | 2091796  | 1359535   |
| Polish     | pl   | Slavic          | 1153817  | 1219494   |
| Portuguese | pt   | Romance         | 2971009  | 1107798   |
| Romanian   | ro   | Romance         | 1960236  | 348036    |
| Russian    | ru   | Slavic          | 6579113  | 1939546   |
| Swahili    | sw   | Niger-Congo     | 11878    | 34107     |
| Telugu     | te   | Dravidian       | 7155     | 80131     |
| Thai       | th   | Kra-Dai         | 5412     | 156505    |
| Turkish    | tr   | Turkic          | 3524089  | 353028    |
| Urdu       | ur   | Indo-Iranian    | 154912   | 96773     |
| Vietnamese | vi   | Austro-Asiatic  | 1019445  | 566375    |
| Chinese    | zh   | Sino-Tibetan    | 434378   | 1027950   |

Table 8:  Number of documents per language used for pre-training. Languages represent a range of families and geographical regions. The Germanic, Hellenic, Romance, Slavic, and Indo-Iranian families are part of a broader Indo-European family.

| Language | Source | Zero-shot Unsupervised Output |
|---|---|---|
| French | Katherine Coleman Goble Johnson, née le 26 août 1918 à White Sulphur Springs (Virginie-Occidentale) et morte le 24 février 2020 à Newport News (Virginie), est une physicienne, mathématicienne et ingénieure spatiale américaine.Elle contribue aux programmes aéronautiques et spatiaux du National Advisory Committee for Aeronautics (NACA) puis de la National Aeronautics and Space Administration (NASA). | Katherine Coleman Goble Johnson (August 26, 1918 – February 24, 2020) was an American physicist, mathematician, and space engineer. She contributed to the aeronautics and space programs of the National Advisory Committee for Aeronautics (NACA) and later the National Aeronautics and Space Administration (NASA). |
| Spanish | Katherine Coleman Goble Johnson (White Sulphur Springs, Virginia Occidental; 26 de agosto de 1918 - Newport News, Virginia; 24 de febrero de 2020) fue una física, científica espacial y matemática estadounidense que contribuyó a la aeronáutica de los Estados Unidos y sus programas espaciales con la aplicación temprana de las computadoras electrónicas digitales en la NASA. Conocida por su precisión en la navegación astronómica, calculó la trayectoria para el Proyecto Mercury y el vuelo del Apolo 11 a la Luna en 1969. | Katherine Coleman Goble Johnson (August 26, 1918 – February 24, 2020) was an American physicist, space scientist, and mathematician who contributed to the United States aeronautics and its space programs with the early application of digital electronic computers at NASA. Known for her accuracy in astronomical navigation, she calculated the trajectory for Project Mercury and the Apollo 11 flight to the Moon in 1969. |
| German | Katherine G. Johnson (gebürtig Coleman, zwischenzeitlich verheiratet Goble; * 26. August 1918 in White Sulphur Springs, West Virginia; † 24. Februar 2020 in Newport News, Virginia) war eine US-amerikanische Mathematikerin afroamerikanischer Abstammung. Für ihre Beiträge zur Berechnung der Flugbahnen für das Mercury-Programm und den ersten bemannten Flug zum Mond im Rahmen der Apollo-11-Mission wurde sie Ende 2015 mit der Presidential Medal of Freedom ausgezeichnet. | Katherine G. Johnson (née Coleman; August 26, 1918 – February 24, 2020) was an American mathematician. She was awarded the Presidential Medal of Freedom in 2015 for her contributions to the calculation of the flight paths for the Mercury program and the first manned flight to the Moon in the Apollo 11 mission. |
| Italian | Katherine Coleman Goble Johnson (White Sulphur Springs, 26 agosto 1918 – Hampton, 24 febbraio 2020) è stata una matematica, informatica e fisica statunitense. Contribuì alla scienza dell'aeronautica statunitense e ai programmi spaziali, già dal primo utilizzo dei computer elettronici digitali da parte della NASA. Venne molto apprezzata per l'accuratezza che poneva nel calcolo della navigazione spaziale computerizzata e per il lavoro tecnico dirigenziale pluridecennale svolto alla NASA: da quando calcolava le traiettorie delle orbite, paraboliche e iperboliche, le finestre di lancio e i percorsi di ritorno di emergenza per molti voli, al Project Mercury, incluse le prime missioni NASA di John Glenn, Alan Shepard, le traiettorie di inserzione lunare nei voli lunari del programma Apollo, continuando con il lavoro sul programma dello Space Shuttle, infine con la progettazione dei primi piani per la missione su Marte. | Katherine Coleman Goble Johnson (White Sulphur Springs, August 26, 1918 – Hampton, February 24, 2020) was an American mathematician, computer scientist, and physicist. She contributed to the science of the U.S. Air Force and space programs, as early as the first use of digital electronic computers by NASA. She was highly regarded for the accuracy she put into computerized space navigation calculations and for the decades-long technical leadership work she performed at NASA: from calculating orbital trajectories, parabolic and hyperbolic, launch windows, and emergency return paths for many flights, to Project Mercury, including the first NASA missions of John Glenn, Alan Shepard, lunar insertion trajectories in the Apollo lunar flights, continuing work on the Space Shuttle program, and finally designing the initial plans for the Mars mission. |

Table 9: Example zero-shot unsupervised inputs and outputs (truncated for clarity).