[Reviews · NeurIPS 2020]

Review 1

Summary and Contributions: This work proposes a novel pre-training approach for NLP applications - utilizing reconstruction of a document from relevant retrieved documents instead of the more traditional MLM or LM style objectives. All documents in the pre-training set are first sharded based on available meta-data into shards of related documents. From each shard, given a target document, a set of relevant documents is extracted based on similarity of the document representations extracted from the encoder. These relevant documents are then fed as evidence to a seq2seq model to reconstruct the target document. This pre-trained model is then evaluated on a variety of NLP tasks, including "unsupervised" MT, semi-supervised MT, Summarization, QA and cross-lingual retrieval.

Strengths: 1. Very interesting and novel approach to pre-train seq2seq models on reconstruction from relevant documents instead of the mainstream MLM/LM based objectives. 2. Given that the pre-training objective is now closer to downstream generative tasks, the authors observe strong empirical results on a variety of NLP tasks. 3. The details of the approach are clearly described and the paper is easy to follow.

Weaknesses: 1. It's not clear how important the details of the batch construction (Section 2.4) are to the eventual performance of the proposed approach. For example, when pre-training on Wikipedia, documents on the same topic in different languages are put in the same shard, imposing an indirect supervision towards the translation task. Similarly, when pre-training on News, only the first chunk of each document can be treated as target (Section 3, Data Pre-processing), again imposing a preference towards summarization. The consequences of these choices are not clear due to a lack of ablation studies. 2. The choice of some pre-training hyper-parameters (described in Section 3 pre-training) is not very well motivated. Does training twice on CC-News with varying # workers and a cyclical rate perform better than single-stage pre-training? 3. While the pre-training objective might play a role in the performance improvements observed here, MARGE is a relatively larger model with a different architecture from mBART / XLM-R. Ablating the effect of these architectural choices might convey more information regarding the effectiveness of the pre-training objective.

Correctness: No major issues except those listed in weaknesses above.

Clarity: Yes, details of the approach and experiments are clearly described and the paper is easy to follow.

Relation to Prior Work: Yes, prior work is adequately discussed.

Reproducibility: Yes

Additional Feedback:


Review 2

Summary and Contributions: The paper proposes a novel pre-training multi-lingual multi-document document paraphrasing objective. Given a document the model scores/retrieves relevant documents that are used to generate the first document. The model model is trained from random initialization; uses self-attention across multiple documents weighted by the relevance scores. (The model however uses simple heuristics based on document metadata to construct related document shards that are then scored and used for batch construction.) The model is evaluated on a diverse set of multi-lingual tasks and compared with recent multi-lingual pre-trained models. The model seems to be more data efficient than the previous work. The model does well on cross-lingual sentence retrieval, summarization, paraphrasing, competitively on MLQA (but more data efficient), similarly to mBART on supervised document MT.

Strengths: Novel model and pre-training objective, doing well on a diverse set of tasks.

Weaknesses: - The data used seems to be different to the benchmarks, and in particular, the language/document distribution (compared to data for the benchmarks) is unclear from the paper (and in particular how this could affect performance on different tasks). - The unsupervised document MT doesn't seem to have any baselines/benchmarks.

Correctness: Yes.

Clarity: Generally well written. Please clarify (in the paper): - How do you control the output language of document MT? - It's hard to go from a benchmark name to a citation by searching the paper, e.g. for XLM-R. (e.g. add a citation next to first mention of the name or the model name at the first citation)

Relation to Prior Work: Related work is cited; could be discussed in more detail.

Reproducibility: Yes

Additional Feedback:


Review 3

Summary and Contributions: This paper presents a pre-training method for sequence-to-sequence models based on multilingual document retrieval. Instead of masking parts of text, the model is pretrained like an auto-encoder where the bottleneck is a collection of documents relevant to the input document. The weights given to the relevant documents are computed by a Transformer-based encoder, and the likelihood of the input document is computed by a Transformer-based sequence-to-sequence model. This sequence-to-sequence model can then be fine-tuned to be applied to various multi-lingual tasks. The authors have conducted several multi-lingual tasks and shown that their approach compares favorably to state-of-the-art pretraining models.

Strengths: This is a novel pretraining approach that is qualitatively different from existing masking-based methods. The pretrained model is especially good at cross-lingual sentence retrieval.

Weaknesses: Some of the experimental details are not very clear. Researchers would not be able to reproduce the reported results even if they had enough computational resources. What are the values of M and k? How many documents are there in a shard? Is there any plan to release the code? I am also wondering how the authors made sure that there was no overlap between the datasets used in evaluation (e.g., IWSLT2017 and WMT19) and the data used for pretraining (i.e., CC-NEWS).

Correctness: The CC-NEWS corpus used in Liu et al. (2019) seems to be a monolingual corpus. How can it be used to pretrain the proposed model? How was the model used for summarization and paraphrasing? Was it fine-tuned for each task?

Clarity: The paper is mostly well written and easy to read, but there are some descriptions that are not very clear. The Equation (4) seems to contain a misplaced equal sign. Line 48: z_i should probably be z_j Table 1: Why is MARGE-NEWS not listed here? Line 225: Are the inputs not included in the dataset used for pretraining?

Relation to Prior Work: Relationships between the proposed method and relevant papers are discussed.

Reproducibility: No

Additional Feedback: Figure 1: f(x, z_j): the parentheses should not be italicized. Line 6: We show -> We show that Line 23: sequence to sequence -> sequence-to-sequence Line 29: documents may -> document may Line 50: seq2seq -> sequence-to-sequence Line 137: using the -> using? Line 179: model encoder -> encoder? Line 174: mBERT, XLM and mBART -> mBERT [Reference], …


Review 4

Summary and Contributions: This paper modifies the Transformer encoder-decoder model with the purpose of learning a pre-trained language model through document retrieving and reconstruction. The proposed model shows strong performance on a range of discriminative and generative tasks in many languages.

Strengths: The model achieved strong performances in many tasks. The method allows effective zero-shot learning in many cases, which is interesting to learn retrieval and reconstruction jointly.

Weaknesses: The overall idea of retrieving related texts for pre-training is similar to REALM. The adopted retrieval method needs more refinement in detail. It is roughly based on overall document cosine similarity, which may involve much noise. Besides, the retrieval task is latently problematic, as it is closely related to the training target of the model and is not capable of reflecting the effectiveness of the encoder. The machine translation also does not measure the performance of the encoder because it relies on the decoder to generate the correct target sequence. The experiments and experimental settings are insufficient. Actually ablation studies are absent, readers may have to ask a series of basic questions such as how the hyper-parameters are selected? How the threshold k of similar documents was set? The model is expected to be an alternative to the current MLM models, but the authors do not provide crucial experiments that compare the performance of their method and other MLM models on any tasks. Therefore, the overall evaluation is not convincing enough when there is no explicit evidence are given to show the encoder is indeed strong, especially, the performance on multi-language machine translation tasks are not considerable. The decoder has 4 additional Transformer layers, which is expected to capture local information of the target sequence as illustrated in section 3. However, the experimental effect of this additional module is not validated in the following sections. The model is expected to have strong performance on both discriminative and generative tasks in many languages, while the experiments mainly focus on generation tasks such as machine translation. More experiments on discriminative tasks need to be conducted.

Correctness: yes

Clarity: well written though the evaluation part needs serious improvement

Relation to Prior Work: yes

Reproducibility: No

Additional Feedback: Are there any specific rules about which documents are used as target documents?? Is a piece of text allowed to be evidence and target documents at the same time in one batch? Would this affect the model performance since the relevance module would assign higher similarity score to a pair where the evidence and target are identical?

[Meta-Review · NeurIPS 2020]

This paper present a novel pretraining idea and demonstrates strong empirical results on a number of tasks. Right now the paper reads a bit like a system description and it would be good consider adding some ablation experiments to shed some light on the various design choices. This might meant that some of the tasks might need to be relegated to the appendix to create space for these additional ablation experiments. In the eyes of the AC some ablations would be more useful than the current enumeration of tasks. It would be also be good to think about alternative names for describing the MT setup. If I understand the setup correctly, the corpus that your model has access to contains translations for the sentences it is being trained to translate, it just doesn't know the sentence-level alignment. This is less supervision than what 'supervised' systems have (which are trained on sentence-aligned parallel corpora) but more supervision than unsupervised systems that are trained on monolingual corpora. It is therefore not fair to call your approach unsupervised. It seems closer to a transductive setting. One can fairly easily imagine how borrowed words and entity names can enable the model to fairly quickly find some of the parallel sentences and bootstrap its way from there.